# Sol-Gel Processing of Bismuth Germanate Thin-Films

**Mihail Secu, Corina Elisabeta Secu, Teddy Tite and Silviu Polosan ***

National Institute of Materials Physics, P.O. Box MG 7, 077125 Magurele, Romania; msecu@infim.ro (M.S.);
cesecu@infim.ro (C.E.S.); teddy.tite@infim.ro (T.T.)

**\*** Correspondence: silv@infim.ro; Tel.: +40-214-8-268

**Abstract:** This study aims to obtain uniform and homogeneous bismuth germanate oxides thin films by spin coating and using the sol-gel technique with different precursors, followed by low-temperature annealing at 560 °C. By using $Bi(NO_3)_3$ precursors, we have obtained transparent, yellowish thin films with a 200 nm thickness. The structural analysis of the initial sol-gel powder has shown the presence of two crystalline structures, the cubic $Bi_4Ge_3O_{12}$ (BGO) and monoclinic $Bi_2GeO_5$ crystallites, which evolves towards the BGO structure after annealing. The elemental analysis confirmed the composition of the desired compound $Bi_4Ge_3O_{12}$ with 60 wt % $GeO_2$ and 40 wt % $Bi_2O_5$. On the other hand, by changing the precursor to $(Bi(CH_3COO)_2$, the film thickness increased to 500 nm thicker due to the high viscosity of the sol, and a dominant monoclinic $Bi_2GeO_5$ crystalline structure appeared. The elemental analysis revealed a nonstoichiometric composition with 38 wt % $GeO_2$ and 62 wt % $Bi_2O_3$. Due to the low $GeO_2$ phase content that reacted with metastable $Bi_2GeO_5$, we obtained cubic $Bi_4Ge_3O_{12}$ as a secondary phase, with $Bi_2GeO_5$ as a dominant crystalline phase. The redshifts of both absorptions and emissions spectra peaks confirmed a different disorder structure as an interplay between the cubic $Bi_4Ge_3O_{12}$ (BGO) and monoclinic $Bi_2GeO_5$ phases.

**Keywords:** BGO thin films; sol-gel; spin coating deposition; slow thermal annealing

---

## 1. Introduction

Bismuth germanate oxides, especially $Bi_4Ge_3O_{12}$ (BGO), remain on the top of scintillating materials that can be used efficiently as radiation detectors due to some unique properties like high light output and stable and reproducible production [1]. Modern applications in high-energy physics, nuclear medicine, geophysics and today homeland security place more stringent requirements on the energy resolution and efficiency of relatively small detectors. These materials, especially in their crystalline form, are used as imaging calorimeters that measure the energy and trajectory of incident particles and provide effective electron/hadron discrimination based on shower images at the cosmic ray proton spectrum from 40 to 100 TeV [2].

Bismuth germanate has a cubic crystalline structure known as eulityne, and luminescence properties are related to intra-ionic transitions of $Bi^{3+}$ cations from the 6s6p level to the $6s^2$ ground level [3].

Bismuth germanate oxides can be used for gamma radiation [4], but the most striking application is in the medicine area, where BGO crystals are used as absorber detectors of a Compton camera for prompt γ-ray imaging during ion beam therapy [5]. In these applications, a scintillator is a streaked block of bismuth germanate oxide (BGO) providing 8 × 8 pseudo pixels and compared with materials such as $LaBr_3$:Ce and LYSO:Ce detectors. Concerning the detection efficiency, BGO detectors were determined to be comparable to, and better than $LaBr_3$:Ce and LYSO:Ce, both on account of having no intrinsic radioactivity. In the same area, BGO nanoparticles were used in biolabel applications [6].

The preparation methods of BGO materials vary from BGO single crystals [7] by using the Czochralski method, vacuum thermal evaporation [4,8], glasses and glass-ceramic materials [9–13], pressure-assisted

combustion [14], and radio frequency (rf) sputtering [15]. A special case of BGO nanocrystals was successfully synthesized using a room-temperature chemical process under atmospheric pressure in an aqueous solution [16]. These BGO nanocrystals were obtained by changing the pH values of solutions between 1 and 9 with different stirring times. However, each of the abovemetioned methods presents specific technical requirements and difficulties. An alternative option is by using the sol-gel route with its advantages: low processing temperature, the ability to control the purity and homogeneity of the final materials on a molecular level, and the large compositional flexibility [17–19]. Several papers describe the obtaining of bismuth germanate oxides by the sol-gel method, but all of them use the calcination method at 900 °C to obtain a BGO compound [14,20].

The study aims are to obtain bismuth germanate oxides by the sol-gel technique followed by low-temperature annealing without a calcination process. In our case, the sols obtained by using germanium and bismuth precursors were spin-coated on glass substrates and slowly thermally annealed at 560 °C to reach the crystallization of the resulted thin films. This temperature was set in a range of crystallization between 540 and 578 °C, determined from BGO non-isothermal crystallization curves with differential thermal analysis (DTA) [21]. The slow thermal annealing avoided the calcination process at 900 °C, which created BGO structures by a solid-state reaction between two oxides, $Bi_2O_3$ and $GeO_2$ [9] to obtain uniformly and homogeneously deposited thin films that can be used as detectors.

## 2. Materials and Methods

*Sol-Gel Synthesis of Bismuth Germanate Oxides*

For the preparation of bismuth germanate oxides, we used the sol-gel method with Germanium(IV) ethoxide (purity: 97%), $Ge(OC_2H_5)_4$ (Alfa Aesar), citric acid (purity: >99%), $C_6H_8O_7$ (Alfa Aesar), ethyl alcohol (purity: >99.8%; Sigma-Aldrich), Bismuth(III) nitrate ($Bi(NO_3)_3$)•$5H_2O$) 98%, Bismuth(III) acetate (purity: 99.99%), $Bi(CH_3COO)_2$ (Alfa Aesar), and glacial acetic acid (purity: 99.7%; $CH_3COOH$, Alfa Aesar) as starting materials. In a typical synthesis, aqueous metal salts are mixed with a proper acid to form a liquid sol. The chemical reactions can be formally described by the following three equations ([17] and references therein):

$$\text{Hydrolysis: } \equiv \text{Ge-OR} + \text{H}_2\text{O} \rightarrow \equiv \text{Ge-OH} + \text{R-OH,}$$

$$\text{Alcohol condensation: } \equiv \text{Ge-OH} + \text{RO-Ge} \rightarrow \equiv \text{Ge-O-Ge} \equiv + \text{R-OH,} \tag{1}$$

$$\text{Water condensation: } \equiv \text{Ge-OH} + \text{HO-Ge} \rightarrow \equiv \text{Ge-O-Ge} \equiv + \text{H}_2\text{O with R = -CH}_2\text{-CH}_3.$$

For the first synthesis, prepared by a bismuth nitrate precursor, we mixed 0.1 g citric acid, 0.8 mL water, 1.0 mL absolute ethyl alcohol, and 1.0 mL Germanium(IV) ethoxide into a round flask with a plug at room temperature and under continuous magnetic stirring [22]; citric acid was used here as an effective chelating agent. Then, 2.9163 g of stoichiometric Bismuth(III) nitrate pentahydrate was dissolved in the 5.0 mL glacial acetic acid to prevent hydrolysis of $Bi^{3+}$, and the solution was slowly dropped to the prehydrolyzed solution under constant stirring.

For the second synthesis, prepared by a bismuth acetate precursor, Germanium(IV) ethoxide diluted with an equal volume of ethyl alcohol was hydrolyzed under constant stirring, and a solution was obtained. The volume ratio of $Ge(OC_2H_5)_4$:$H_2O$:$CH_3COOH$ was 7:2:1, and glacial acetic acid was used here as a catalyst. Then, the required amount of Bismuth(III) acetate was dissolved in acetic acid with a molar ratio of 1:14, and a second solution was obtained. This solution was dropwise added into the first solution, followed by stirring for 30 min at room temperature.

For each of the synthese, a milky white sol was obtained that was used for depositing 250 μL of each solution by spin coating with an Ossila spin-coater with the following parameters: 2000 rpm for 30 s on glass substrates. Both samples with the deposited thin films and solutions were dried at 200 °C for 1 hour. The thin films were annealed using a software-controlled furnace Nabertherm at a speed of 3 °C/min, until the temperature reached 560 °C. Then, they were kept for 1 hour and naturally cooled

at room temperature. The thicknesses of the obtained films were evaluated by using a Thetametrisis equipment for precise and nondestructive characterization of transparent and semitransparent single films or stacked films. This ellipsometric equipment was used to perform reflectance in the 370–1020 nm spectral range and measure the thicknesses from 12 up to 90 μm. The optical constants for BGO (major phase) thin films were obtained by classical ellipsometry performed with a Woollam Variable Angle Spectroscopic Ellipsometer (VASE) system, equipped with a high-pressure Xe discharge lamp incorporated in an HS-190 monochromator [23].

XRD measurements were performed on a Bruker D8 Advance type X-ray diffractometer with a copper target X-ray tube and a LynxEye one-dimensional detector. The X-ray tube was operated at 40 kV and 40 mA. $CuK\alpha_1$ radiation ($\lambda = 1.54056$ Å) was used as an X-ray source. The Crystal Sleuth software was used for the analysis of XRD data.

Absorption and emission spectra were recorded with a Jasco 815 spectrometer while the SEM was performed with a Zeiss EVO 50 scanning electron microscope with a $LaB_6$ cathode and a Brucker EDX system. Based on the SEM images, the equipment may generate secondary electrons, backscattered electrons, characteristic X-rays, and cathodoluminescence.

Micro-Raman spectra were recorded at room temperature in a backscattering configuration using a LabRAM HR Evolution spectrometer (Horiba Jobin-Yvon) equipped with a confocal microscope. A He–Ne laser operating at 633 nm was focused on the surface of sol-gel glass powders or films with an Olympus 100× objective. The spectrum of a reference BGO crystal was recorded by using an Olympus 10× objective. Accurate calibration was performed by checking the Rayleigh and Si bands at 0 and 520.7 $cm^{-1}$, respectively. The laser excitation power was adjusted to avoid laser-induced heating. The scattered Raman signal was then recorded by using a confocal spectrometer, dispersed through an 1800 lines/mm diffraction grating and detected by using a CCD camera. The spectral resolution of the reported spectra was around 0.5 $cm^{-1}$.

The bonding architecture of glass powders and films was investigated by FTIR with a Perkin Elmer Spectrum BX II spectrophotometer. The spectra were recorded in attenuated total reflectance (ATR) mode (PikeMiracle head) in the range of 500–4000 $cm^{-1}$, with a resolution of 4 $cm^{-1}$ and a total of 32 scans for each sample.

## 3. Results

The properties of the obtained bismuth germanate thin films were correlated with the quality of the sol precursors that had different viscosities and homogeneities. For example, the sample prepared by a bismuth nitrate precursor became more homogeneous, and the obtained thin film was around 200 nm thick due to a lower viscosity compare with that of the sample prepared by a bismuth acetate precursor, which was thicker and had a thickness of around 500 nm (Figure 1). This thickness result also showed the sample prepared by the bismuth nitrate precursor had good transparency and the sample prepared by the bismuth acetate precursor was opaque.

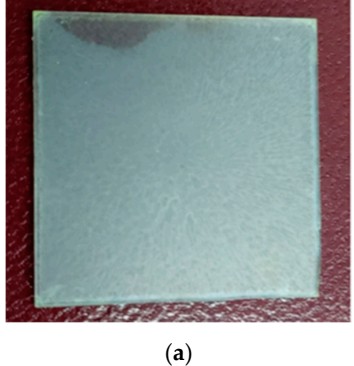 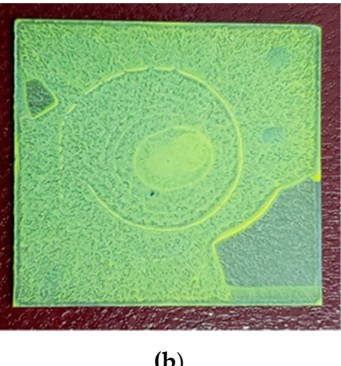

(**a**)　　　　　　　　　　　　　　　　(**b**)

**Figure 1.** Photos of the two thin films samples (2.54 × 2.54 cm) prepared by spin coating and annealing at 560 °C: (**a**) sample prepared with bismuth nitrate; (**b**) sample prepared with bismuth acetate.

### 3.1. Structural Analysis by XRD

The structural analysis of bismuth germanate oxides obtained by the sol-gel method involved the comparisons of the crystallization processes and elemental analyses in the obtained sol-gel powder after the drying procedure at room temperature and the thermally annealed powder and thin films at 560 °C.

The crystallization features were studied by XRD, and the comparison was done between the two samples (Figure 2a,b)). Different types of bismuth germanate structures appeared, but the main two structures were cubic $Bi_4Ge_3O_{12}$ (BGO) and monoclinic $Bi_2GeO_5$.

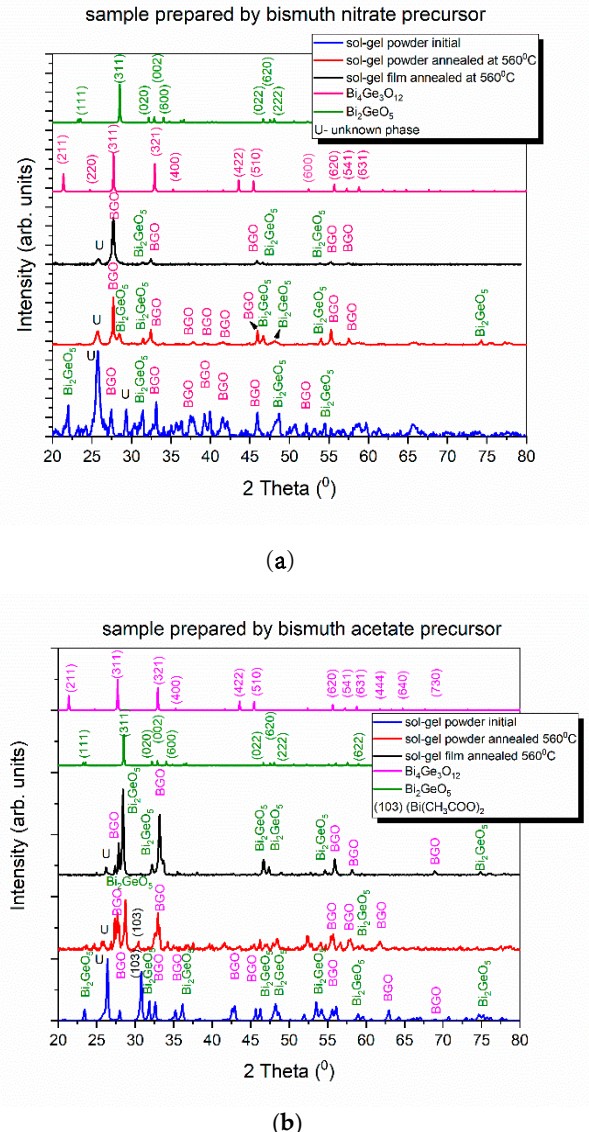

**Figure 2.** (**a**) XRD patterns for the sample prepared by bismuth nitrate. (**b**) XRD patterns for the sample prepared by bismuth acetate.

In the sample prepared by the bismuth nitrate precursor, even in the as-prepared sol-gel powder, some peaks of BGO can be identified, especially (310) at 27.85°, (321) at 31.9°, and (400) at 34.12°. These three peaks defined a cubic structure with a = b = c = 13.22 Å and α = β = γ = 90°. The as-prepared powder had lower crystallinity, but after thermal annealing at 56 °C for 1 hour, two crystalline structures $Bi_4Ge_3O_{12}$ (BGO) and $Bi_2GeO_5$ were observed as major phases. The presence of $Bi_2GeO_5$ was evidenced by the (311) reflection peak at 28.53°, but the dominant crystalline structure was BGO, for which the

(310) reflection at 26.85° significantly increased after annealing. The lattice parameters of the cubic structure (a = b = c = 10.8 Å) were close to the value of 10.52 Å for the BGO monocrystal [24]. In the annealed thin film, the BGO crystalline structure was the dominant one revealed by the (310) peak, consistent with a cubic structure with a = b = c = 10.48 Å. The crystallite dimensions were calculated based on the Debye Scherrer equation, assuming there was no stress action on the nanocrystallites. Thus, in the initial sol-gel powder, the dimension was estimated as about 20 nm, which rapidly increased in the thermally annealed powder to 48 nm and remained somewhere at around 25 nm in the annealed thin film.

Similar results were obtained in the sample prepared by the bismuth acetate precursor, where the intensity of the XRD peaks increased because the film thickness was higher; the BGO structure was revealed by the peak at 27.9° assigned to the (310) reflection. However, after thermal annealing, the $Bi_2GeO_5$ crystalline phase seemed to be the dominant one, evidenced by the strong peak at 28.5° and accompanied by the BGO crystalline phase marked with two main peaks at 27.65° and 33.9° assigned to the reflections of (310) and (400) plans. The $Bi_2GeO_5$ peaks were enhanced by the thermal annealing of the thin film sample compared with those in the BGO structure. Moreover, the nanocrystallites size slightly increased from 36 nm in the initial powder to 40 nm in the annealed powder and then 48 nm in the annealed thin film, only taking into account BGO peaks. The cubic structure of the thin film was consistent with the lattice parameters values of a = b = c = 10.61, which were close to the values of the BGO crystal. In both samples, a peak centered at 26.3° appeared, which was not identified, probably related to germanium ethoxide. This peak disappeared after annealing in both samples. The (103) peak of bismuth acetate can be seen at 30.8°.

### 3.2. Structural Analysis by Elemental Analysis with the Energy-Dispersive X-ray Spectroscopy (EDS) Method

The elemental analysis performed by using the EDS method enabled the determination of composition and the ratio between bismuth and germanium oxides. The procedure was applied to the initial sol-gel powder, assuming the same elemental compositions after subsequent thermal annealing. The EDX patterns of the sample prepared by the bismuth nitrate precursor are given in Figure 3, together with the elemental analysis in Table 1.

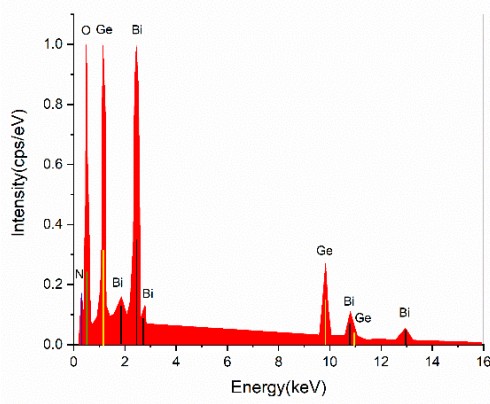

**Figure 3.** Elemental analysis of the sample prepared by the bismuth nitrate precursor.

**Table 1.** Atomic concentrations of the main elements in the sample prepared by the bismuth nitrate precursor.

| Element | Atomic Number | Atomic Concentration (wt %) | Error (%) |
|---|---|---|---|
| O | 8 | 85.92 | 5.5 |
| Ge | 32 | 8.56 | 0.6 |
| Bi | 83 | 5.52 | 1.2 |
| Total | | 100 | |

The atomic concentration of germanium was 8.56 wt %, while the bismuth concentration was 5.52 wt %, which resulted in 60.8 wt % $GeO_2$ and 39.2 wt % $Bi_2O_5$ in the initial powder, preserving the desired composition of 60 wt % $GeO_2$ and 40 wt % $Bi_2O_5$ from the chemical synthesis. The EDX patterns of the sample prepared by the bismuth acetate precursor are given in Figure 4, together with the elemental analysis in Table 2.

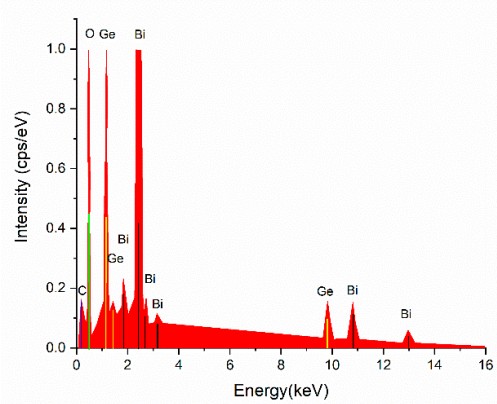

**Figure 4.** Elemental analysis of the sample prepared by the bismuth acetate precursor.

**Table 2.** Atomic concentrations of the main elements in the sample prepared by the bismuth acetate precursor.

| Element | Atomic Number | Atomic Concentration (wt %) | Error (%) |
|---------|:-------------:|:---------------------------:|:---------:|
| O       | 8             | 88.82                       | 5.4       |
| Ge      | 32            | 4.25                        | 0.3       |
| Bi      | 83            | 6.93                        | 1.4       |
| Total   |               | 100                         |           |

In this case, the atomic concentration of germanium was 4.25 wt %, while the bismuth concentration was 6.93 wt %, which resulted in 38 wt % $GeO_2$ and 62 wt % $Bi_2O_5$ in the initial powder.

### 3.3. Optical Spectroscopy

The optical absorption spectrum (Figure 5a) of the sample prepared by the bismuth nitrate precursor with a thickness of around 200 nm consisted of two absorption peaks, one broadband centered at 595 nm and a sharper one at 320 nm. For the sample prepared by the bismuth acetate precursor, the absorption spectrum was not recorded, because it was not transparent. Concerning the emission spectra of both samples excited at 320 nm, the photoluminescence signal appeared more intense at 520 nm in the first sample and less intense at 553 nm in the second sample. The absorptions and emissions were assigned in comparison with those of BGO glasses and crystals [23].

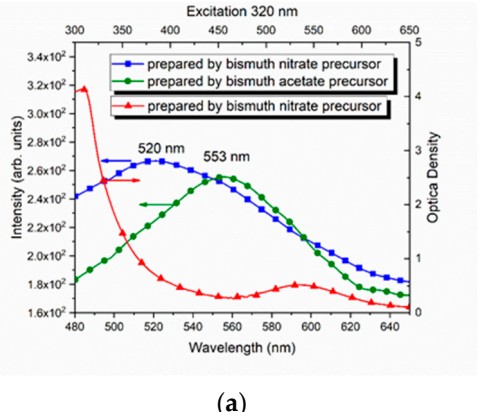

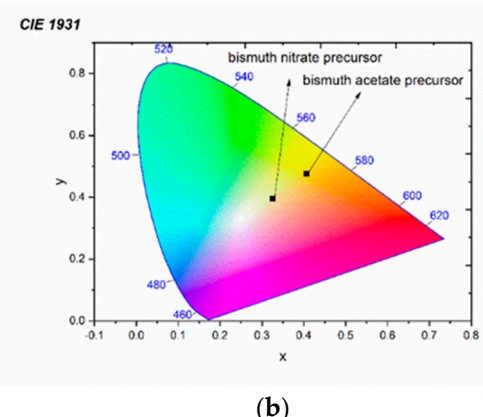

(**a**)                                                                            (**b**)

**Figure 5.** (**a**) Optical absorption spectrum of the sample prepared by the bismuth nitrate precursor and emissions spectra of the samples prepared by the bismuth nitrate and bismuth acetate precursors. (**b**) Reflectance colors of the samples prepared by the bismuth nitrate and bismuth acetate precursors on the CIE 1931 chromaticity map.

From the optical reflectance spectra, we obtained the chromaticity of the annealed thin films (Figure 5b). The thinner sample prepared by the bismuth nitrate precursor exhibited a yellow color, suggesting a predominant single phase compared with the thicker one (sample prepared by the bismuth acetate precursor), which revealed a redshift towards the yellow-orange color on the CIE 1931 chromaticity map.

All the peak vibrations identified in the Raman spectra for the initial sample, the powder annealed at 560 °C, and the annealed thin films were compared with those shown in the BGO crystal (Figure 6a). In the sample prepared by the bismuth nitrate precursor, the first observed peak at 63.8 cm$^{-1}$ in the BGO crystal was broadened and shifted to 73 cm$^{-1}$ in the initial powder and 72 cm$^{-1}$ in the annealed powder. For the annealed film, this vibration appeared at 71 cm$^{-1}$ and was assigned to the lattice-mode vibration in BGO. The totally symmetric vibrational mode A located at 88.5 cm$^{-1}$ was observed in the sample prepared by the bismuth nitrate precursor and was gradually decreased in the initial powder, then appeared in the annealed sample and later in the thin film. The next peak vibration at 121.3 cm$^{-1}$ in the initial powder was shifted to be at 124 cm$^{-1}$ in the annealed powder and thin film, while in the BGO crystal it appeared at 122.5 cm$^{-1}$. This peak had a strong intensity in the initial sample, and its half width decreased in the annealed powder and thin film. The most striking mode was at 315.5 cm$^{-1}$ that appeared only in the annealed powder and film and was shifted to be at 360 cm$^{-1}$ in the BGO crystal. The peak was related to the total symmetric mode of the Bi–O bond stretching. The peaks from above 410 up to 825 cm$^{-1}$ were assigned to GeO$_4$ modes and decreased from the initial sol-gel powder to the annealed thin film and powder.

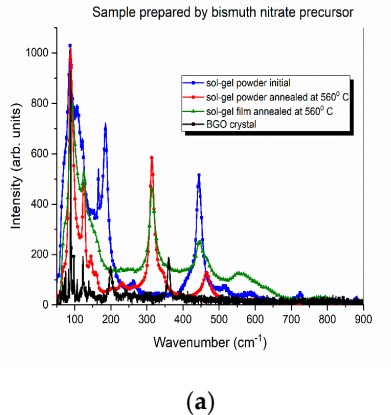

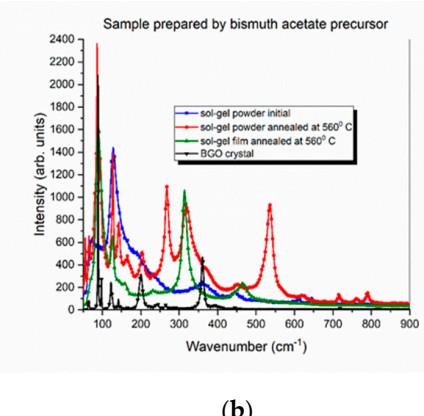

(**a**)                                                                            (**b**)

**Figure 6.** (**a**) Raman spectra of the sample prepared by the bismuth nitrate precursor; (**b**) Raman spectra of the sample prepared by the bismuth citrate precursor.

In the sample prepared by the bismuth citrate precursor (Figure 6b), the peaks was similar to those in the sample prepared by the bismuth acetate precursor, except at 269 cm$^{-1}$, which was quite intense in the sol-gel annealed powder and cannot be seen in the annealed thin film.

The FTIR spectra recorded on both the annealed powder and thin film are depicted in Figure 7a,b. In the sample prepared by the bismuth nitrate precursor, two main peaks can be observed at 553 and 584 cm$^{-1}$ in the initial sol-gel powder and shifted to 548 and 589 cm$^{-1}$ in the sol-gel powder and film annealed at 560 °C. Less visible were the next two peaks at 742 and 778 cm$^{-1}$ that can be seen not only in the annealed powder, but also in the annealed thin film as one broadband (Figure 7a).

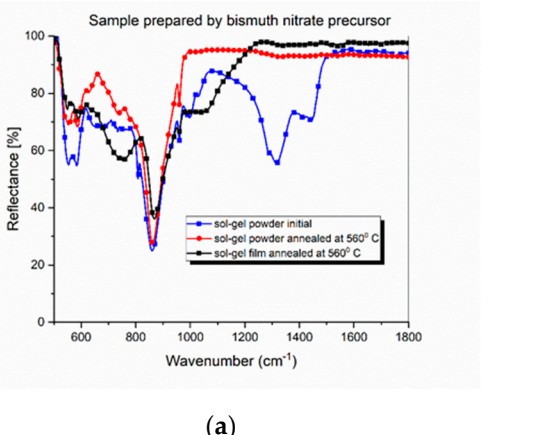 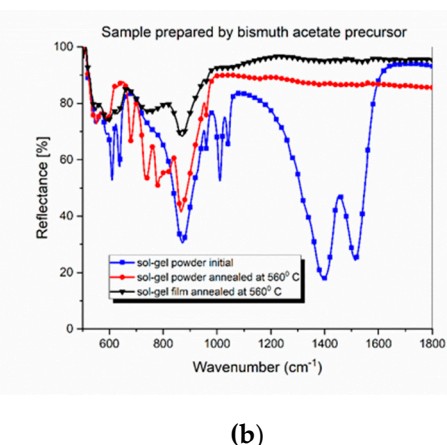

(**a**)                                                    (**b**)

**Figure 7.** (**a**) FTIR spectra of the sample prepared by the bismuth nitrate precursor; (**b**) FTIR spectra of the sample prepared by the bismuth acetate precursor.

The most intense peak can be seen at 861 cm$^{-1}$ in the initial sol-gel powder and shifted to 868 cm$^{-1}$ in the annealed thin film. The last group of peaks that arose from the bismuth germanate structure appeared at 940 and 998 cm$^{-1}$, and the other ones, which appeared at 1320 and 1425 cm$^{-1}$, were assigned to organic residues, because they disappeared after thermal annealing.

Similar assignments can be made for the sample prepared by the bismuth acetate precursor (Figure 7b), but the peaks were much better resolved due to the higher thickness of the obtained film.

The optical microscopy image of the sample prepared by the bismuth nitrate precursor taken during the Raman measurements revealed a smooth, homogeneous and uniform distribution of the BGO thin film after slow thermal annealing (Figure 8).

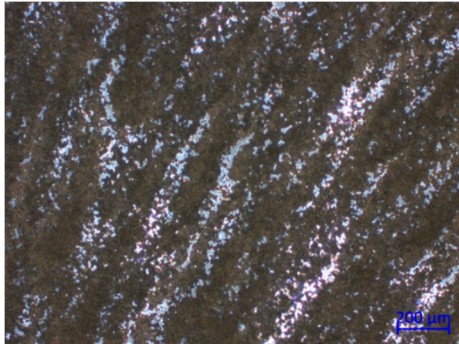

**Figure 8.** The optical image of the sample prepared by the bismuth nitrate precursor.

## 4. Discussion

The annealing procedure at 560 °C was performed in the region of the BGO crystallization peak, determined for the glass samples obtained by the melt quenching procedure at 556 °C (829 K) [21]. In fact, the annealing point was overlapped with the second-phase Bi$_2$GeO$_{12}$ crystallization peak

determined by DTA at 568 °C (840 K). Therefore, above this temperature, the bulk crystallization of the $Bi_4Ge_3O_{12}$ (BGO) phase was completed by the formation of intermediary-metastable-phase $Bi_2GeO_5$.

The structural analyses of the initial sol-gel powder and the thermally annealed powder and thin film with a thickness of 200 nm confirmed the coexistence of two crystalline phases before annealing, cubic $Bi_4Ge_3O_{12}$ (BGO) and monoclinic $Bi_2GeO_5$ crystallites, which evolved towards a BGO crystalline structure in the annealed thin film. The most dominant structure seemed to be the BGO crystalline phase, evidenced by the (310) peak, which was consistent with a cubic structure having a = b = c = 10.48, close to the crystallography of BGO monocrystals. The elemental analysis of this thin sample prepared by the bismuth nitrate precursor confirmed the composition of the desired $Bi_4Ge_3O_{12}$ crystalline compound with 60 wt % $GeO_2$ and 40 wt % $Bi_2O_5$. In the case of the sample prepared by the bismuth acetate precursor, with higher thickness, we observed the coexistence of the same cubic $Bi_4Ge_3O_{12}$ (BGO) and monoclinic $Bi_2GeO_5$ crystallites, even after thermal annealing. This fact was marked by the elemental analysis, in which $Bi_2O_3$ was predominant in comparison with the $GeO_4$ structures in the second sample, where 38 wt % $GeO_2$ and 62 wt % $Bi_2O_3$ co-existed. The thermal annealing of 40 wt % $Bi_2O_3$ and 60 wt % $GeO_2$ at 556 °C, produced both phases, cubic and monoclinic, but the monoclinic one was metastable. The reaction between $GeO_2$ and $Bi_2GeO_5$ led to $Bi_4Ge_3O_{12}$ (BGO), shown as following [21]:

$$2Bi_2GeO_5 + GeO_2 \rightarrow Bi_4Ge_3O_{12} \qquad (2)$$

However, according to the elemental analysis, the amount of $GeO_2$ was much lower in the sample prepared by the bismuth acetate precursor, compared with that of $Bi_2O_5$, and the above reaction produced BGO as the secondary phase with the dominant one being the monoclinic $Bi_2GeO_5$ phase.

The broad peak from 595 nm from the absorption spectrum was assigned to the Mie scattering on small crystallites with dimensions comparable with the wavelength of the light scattering. More than that, the excitation into this band did not give photoluminescence. A possible explanation might be related to the small thickness of the film and the small contact angle during the spin coating procedure (also connected with the viscosity of the solution) and the glass substrate, which increased the nucleation processes [24]. This smaller contact angle induced lower surface energy, increasing the nucleation rate. Because the $GeO_4$ tetrahedra still existed in solutions, as can be seen in the vibrational spectra (Figure 7), the nucleation rate should be assigned to the clusterization of these $GeO_4$ tetrahedra, inducing a scattering of light [25]. The explanation is based on the turbidity effect proportional to the size parameter of the cluster, which took into account the refractive index of the glass [26].

The second absorption band centered at 320 nm was similar to the absorption band centered at 290 nm in BGO crystals, which was assigned to the $^3P_1-^1S_0$ transition of $Bi^{3+}$ ions. In the case of the sample prepared by the bismuth nitrate precursor, the absorption at 290 nm in BGO crystals was shifted to 320 nm similar to the previous bismuth germanate melt-quenched samples. In fact, this 320 nm absorption band was not given by the radiation absorption of $Bi^{3+}$ ions [27]. The oxygen atoms, around the $Bi^{3+}$ ions, absorbed the energy on *p*-electrons and transferred it to Bi *p*-electrons. Since the annealed film was formed from two networks $Bi_2O_3$ and $GeO_2$, there were $GeO_4$ tetrahedra that absorbed the radiation energy and transferred it to $Bi^{3+}$ ions [28]. The redshifts of both absorptions and emissions spectra were consistent with the disordered structure that appeared between the cubic $Bi_4Ge_3O_{12}$ (BGO) and monoclinic $Bi_2GeO_5$ phases.

The pure $Bi_4Ge_3O_{12}$ single crystal belonged to the $I\overline{4}3m$ space group with four formula units in crystallographic unit cells [29]. Each unit cell contained 38 symmetrically nonequivalent atoms with respect to the inner vectors of translations: 6 Ge, 8 Bi, and 24 O atoms. More relevant was the existence of isolated $GeO_4$ tetrahedra with $S_4$ symmetry, leading to a deformed octahedral environment of oxygen around $Bi^{3+}$ ions, but these isolated tetrahedra appeared in the amorphous structure during the formation of bismuth germanate thin films. There were 46 vibrational modes, but only 27 were Raman actives and some of them can be compared with those obtained by sol-gel thin film deposition and precursors. The theoretical calculations of Raman-active modes indicated two vibrational modes at 64 and 66 $cm^{-1}$ as transversal (TO) and longitudinal (LO) Bi, O, and Ge vibrations, which can be

seen in the BGO crystal [30]. The totally symmetric vibrational mode A was calculated at 97 cm$^{-1}$ and was assigned as significantly strong peak to the Bi and Ge vibrations, but in the pure BGO crystal appeared at 90 cm$^{-1}$. The presence of this peak at the same wavenumber in all samples and the shift of the first peak suggested fixed positions for Bi and Ge ions and a rearranging mode of oxygen ions in all samples towards a pure BGO structure.

Regarding the theoretical vibration of mode E, it was calculated at 115–126 cm$^{-1}$ and was assigned to the Bi–O–Ge bond bending observed at 123 cm$^{-1}$ in horizontal polarization laser mode and 126 cm$^{-1}$ in vertical mode. For the totally symmetric mode of the $(GeO_4)_3$ against $Bi^{3+}$ bond stretching, the theoretical vibration of mode E was calculated at 315–318 cm$^{-1}$ and was assigned to the transversal mode. However, in the case of the BGO crystal, this vibration showed a quite large shifting. This fact supported the idea of a relaxed BGO structure in thin films and a distorted one in bulk BGO samples. The theoretical calculations for $GeO_4$ vibrations indicated two classes of vibrational modes, assigned to the O–Ge–O bond bending from 407 to 469 cm$^{-1}$ and from 716 to 797 cm$^{-1}$ assigned to the Ge–O bond stretching. In BGO crystals, these modes appeared at 406, 409, and 445 cm$^{-1}$ for the O–Ge–O bond bending and from 703 to 820 cm$^{-1}$ for the Ge–O bond stretching. Similar vibrational peaks were observed in the melt-quenched BGO samples, and their corresponding devitrified the counterparts [12]. For the Raman spectra of the sample prepared by the bismuth acetate precursor, the peaks were more intense, because the sample was thicker than the first sample. Some additional peaks were related to the presence of the bismuth germanate phase, $Bi_2GeO_5$, as can be seen from the XRD patterns (Figure 2). This phase appeared during the crystallization process, but in our case, it appeared in the as-prepared sol-gel powder.

The spectra pattern was consistent with the Raman spectra analysis. A number of 25 active modes were identified from the infrared spectroscopy, but due to the experimental constraints (all spectra were measured in the ATR mode), the low-frequency peaks at 92, 203, and 363 cm$^{-1}$ appeared as overtones or second-order vibrations at 536 and 567 cm$^{-1}$ in the BGO crystals. These vibrations were assigned to $\nu_2$ $GeO_4$ lattice vibrations as the $A_1$ transition type and calculated at 536 cm$^{-1}$ [31]. Another two bands at 740 and 778 cm$^{-1}$ or at 721 and 775 cm$^{-1}$ [32] were assigned to $\nu_1$ and $\nu_3$ vibrations of $GeO_4$ tetrahedra in BGO crystals and are in good agreement with the experimental one at 742 and 778 cm$^{-1}$ in our case. In a similar manner with Bordum, the most intense band appeared at 862 cm$^{-1}$, which was measured at 863 cm$^{-1}$ in the BGO crystals. This band appeared as overtones between 567 and 294 cm$^{-1}$ (not seen in our spectra) with the coupling between two transitions $A_1$ and $F_2$. The 940 and 998 cm$^{-1}$ peaks were assigned to the stretching vibrations of isolated $(GeO_4)^{4-}$ tetrahedra.

## 5. Conclusions

Transparent and homogeneous bismuth germanate thin films were obtained by using the sol-gel method. Their physical properties were strongly influenced by initial raw materials, which induced different viscosities of sol precursors and wettability. This method aimed to avoid the calcination procedure at higher temperatures, which induced a solid-state reaction between oxides and proposed a slow thermal annealing technique for BGO thin-film crystallization. The structure, morphology, and optical properties of the films were influenced by the precursors $Bi(NO_3)_3$ or $Bi(CH_3COO)_2$. The initial sol-gel powder showed the presence of two crystalline structures, the cubic $Bi_4Ge_3O_{12}$ (BGO) and monoclinic $Bi_2GeO_5$ crystallites.

By using bismuth nitrate as a precursor instead of bismuth acetate, the sol-gel processing method led towards the desired BGO compound after annealing, having a dominant phase of thin films deposited on glass substrates and a crystallization point below that of BGO. Thin films with a 200 nm thickness containing cubic $Bi_4Ge_3O_{12}$ particles were obtained by using a $Bi(NO_3)_3$ precursor. Using bismuth acetate as a starting material, the obtained thin films were thicker and less homogeneous and revealed $Bi_2GeO_5$ as a dominant phase but with BGO (around 40%) as a secondary phase after slow thermal annealing.

The optical absorption and luminescence spectra showed redshifts assigned to different disorder structures within the cubic $Bi_4Ge_3O_{12}$ and monoclinic $Bi_2GeO_5$ crystalline phases. The obtaining of the desired BGO compound led to better spectroscopic parameters, especially photoluminescence, which fundamentally contributed to a better scintillating process for high-energy radiation detectors.

The sol-gel procedure of bismuth germanate thin films may induce oxygen defects during crystallization processes, which exhibits a magnetic moment, and hence these thin films with oxygen defects are known as diluted magnetic materials. Similarly, thin films of bismuth germanate glass and glass-ceramic materials can possess such magnetic properties by the sol-gel method as well [33].

**Author Contributions:** Conceptualization, S.P.; methodology S.P.; writing -original draft preparation, S.P.; writing—review and editing, M.S.; synthesis, C.E.S.; investigation, S.P. and T.T. All authors have read and agreed to the published version of the manuscript.

**Funding:** This work was supported by a grant from the Romanian Ministry of Research and Innovation DEXMAV 12PFE/2018 and Core Program PN19-03 (contract no. 21 N/08.02.2019).

**Acknowledgments:** The authors are grateful to Core Program PN19-03 (contract no. 21 N/08.02.2019) and Project DEXMAV (contract no. 12PFE/2018).

**Conflicts of Interest:** The authors declare no conflicts of interest.

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
