# Peer review of "Sol-Gel Processing of Bismuth Germanate Thin-Films"

_coatings, doi:10.3390/coatings10030255_

Round 1

Reviewer 1 Report

Authors have studied the Bismuth germanate film using spin-coating and sol-gel method. They have obtained BGO and monoclinic phase, at least some. Though study is interesting and discussion is good, mnauscript still need to be polished.

  • Could the authors mention what is the purpsoe of citrci acid, triflouric acid ? Are they used as surfactant or reducing agent ? Is it possible to eaplain reaction mechanism? Part of the mechanism is in discussion, but, reaction from the beginning is better.
  • Line 113, how did the authors calculate the thickness of film ? If SEM is used,could the authors put scale in SEM picture, Figure 1.
  • I would suggest to write abbreviation in line 116 because authors have used SG in SRD spectrum. What does asterik represen in Figure 2? In XRD, spetrum, have the authors obtained single BGO and Bi2GeO5 phase because the spetrum for BGO is shown in Figure. This is creating confusion.
  • In EDX, How many spots were used to acquire spectrum ? Is that weighted average ? In line 160, authors have concluded that percentage of Geo2 and Bi2O5 from the concentration of Ge and Bi. Could the atuthors mention, how it is obtained ? Also, EDX spectrum picture is not clear.
  • Does BGO has optical band gap and magnetic behavior ?

Author Response

Dear Reviewer,

                Thank you very much for the careful reading of our manuscript. We have proceeded to answer step-by-step, as following:

Reviewer 1

Authors have studied the Bismuth germanate film using a spin-coating and sol-gel method. They have obtained BGO and monoclinic phase, at least some. Though the study is interesting and discussion is good, the manuscript still needs to be polished.

  • Could the authors mention what is the purpose of citric acid, triflouric acid ? Are they used as surfactant or reducing agent ? Is it possible to explain reaction mechanism? Part of the mechanism is in discussion, but, reaction from the beginning is better.

Answer

The citric acid acts as an effective chelating agent. We have mentioned this thing in the revised version.

The manuscript was completed with the sentence:

 “In a typical synthesis, aqueous metal salts are mixed with the proper acid to form the liquid sol. The chemical reactions can be formally described by the following three equations ([17] and references therein):

 º Ge-OR + H2O ® º Ge-OH + R-OH (hydrolysis)

º Ge-OH + RO-Ge ® º Ge-O-Ge º + R-OH (alcohol condensation)

º Ge-OH + HO-Ge ® º Ge-O-Ge º + H2O (water condensation) (with R = -CH3)

Reviewer 1

  • Line 113, how did the authors calculate the thickness of film ? If SEM is used, could the authors put scale in SEM picture, Figure 1.

Answer

The thickness was evaluated with ellipsometric methods using the Thetametrisis pOrtable equipment in the reflectance mode. The manuscript was completed with the sentence “The thicknesses of the obtained films were evaluated by using Thetametrisis equipment for precise non-destructive characterization of transparent and semi-transparent single films or stack of films. This ellipsometric equipment performs reflectance in the 370-1020nm spectral range and measures the thicknesses between 12 nm up to 90 μm. The optical constants for BGO (major phase) thin films were obtained by classical ellipsometry performed with Woollam Variable Angle Spectroscopic Ellipsometer (VASE) system, equipped with a high-pressure Xe discharge lamp incorporated in an HS-190 monochromator [24].”

Figure 1 is not an SEM image but a real photo of the measured samples of around 1 inch/sq.

Reviewer 1

  • I would suggest to write abbreviation in line 116 because authors have used SG in SRD spectrum. What does asterik represen in Figure 2? In XRD, spetrum, have the authors obtained single BGO and Bi2GeO5 phase because the spectrum for BGO is shown in Figure. This is creating confusion.

Answer

We have removed the asterisks form the figure and text for better understanding, assigning these peaks with BGO structure. Both graphs were re-drawn with theoretical patterns of BGO and Bi2GeO5 and marked in the experimental spectra for better viewing.

Reviewer 1

  • In EDX, How many spots were used to acquire spectrum ? Is that weighted average ? In line 160, authors have concluded that percentage of Geo2 and Bi2O5 from the concentration of Ge and Bi. Could the atuthors mention, how it is obtained ? Also, EDX spectrum picture is not clear.

Answer

The EDX spectra and Raman spectra have been obtained in three to six different points and the Ge and Bi were determined as atomic percentages. The atomic percentage is the number of atoms of that element, at that weight percentage, divided by the total number of atoms in the sample multiplied by 100.

Reviewer 1

  • Does BGO has optical band gap and magnetic behavior?

Answer

Yes. On a previous paper [24] we have measured the bandgap of BGO glasses and glass-ceramic materials by ellipsometry. The obtained band gaps are 4.58 eV (271 nm) for BGO single crystal and 3.6 eV (344 nm), closed to the values obtained by optical absorption, respective 300 nm for the BGO single crystal (determined from the optical spectra) and 350 nm for the BGO glass samples. The results show that the direct band gap value is larger than the indirect band gap. The decreasing of the bang gap in the case of BGO glass samples compared with that of the single crystal shows an increase of the disorder and consequently the more extension of the localized states within the gap according to with the Mott and Davis theory. The 320 nm (3.87 eV) absorption band can be interpreted as disorder oxygen localized states under the conduction band.

And Yes, BGO has magnetic properties. In the paper:

“Ferromagnetic behaviour of bismuth germanate oxides glass-ceramic materials”

Polosan, S ; Negrea, R ; Ciobotaru, IC ; Schinteie, G ; Kuncser, V - JOURNAL OF ALLOYS AND COMPOUNDS, (2015), Volume: 623 Pages: 192-196, DOI: 10.1016/j.jallcom.2014.10.104

In this paper, we have concluded (for the bulk samples): “The content of defects in Bi4Ge3O12(BGO) glass-ceramic materials together with their ordering during crystallization induces ferromagnetic behaviors in these materials. The observed ferromagnetism has to be associated strictly with the GeO4related defects in the atypical amorphous phase.” We have added the reference in the manuscript.

Reviewer 2 Report

I have mentioned my comments and suggestions in the document attached.

Author Response

Dear Reviewer,

            Thank you very much for the careful reading of our manuscript. We have proceeded to answer step-by-step, as following:

Reviewer 2.

The authors have shown a highly interesting and original work. In order for the manuscript to be improved, certain points which are mentioned below should be addressed carefully:

Reviewer 2.

  1. Instead of naming the samples as “Sample 1” and “Sample 2” naming them as “sample prepared by bismuth nitrate precursor” and “sample prepared by bismuth acetate precursor” could provide a more explanatory style within the entirely of the text and captions.

Answer:

The names of both samples were converted in “sample prepared by bismuth nitrate precursor” and “sample prepared by bismuth acetate precursor”, accordingly.

Reviewer 2.

2 There are points to be improved about the XRD analysis study in the manuscript:

o As only the relative intensity values of different patterns are worth comparing when multiple XRD patterns are demonstrated in an image, the exact numerical values are not worth regarding are not shown on the intensity axis typically. So, in Figure 2, the authors should remove the numerical values in the y-axis, and also they should include the interval which includes the full “SG initial” pattern because in its current form in the manuscript the whole pattern is not able to take place as a part of the pattern remains in the below zero region. Besides, keeping more distance between peaks and would provide a more comprehensible and clearer demonstration.

Answer:

The Oy scales were withdrawn as values, while the XRD spectra are arbitrary shifted for a better view.  All initials SG were converted in “sol-gel” for better understanding. The main peaks for Bi4Ge3O12 and Bi2GeO5 were added to the theoretical spectra (obtained from crystallography), while in the experimental patterns, the peaks are marked as “BGO” or “Bi2GeO5”.

Reviewer 2.

o There is a mistake in the notation of planes detected from XRD, both in figure 2 and the text. The planes were shown as [h,k,1] whereas the notation should be like (hkl) in the manuscript. For example, notations like “[3,1,0] reflexion” were used in the entire manuscript were as it should be “(310) reflection”.

Answer:

The [k,k,l] notations were transformed in (hkl) in the manuscript and figures. And the mistake “reflexion” instead of “reflections” was corrected.

o All the peaks from the patterns are not identified and written in Figures 2a and 2b. Instead of mentioning just a few peaks in the reference patterns of Bi4Ge3O12 and Bi2GeO5, making a clear demonstration of the planes of SG initial, SG annealed at 560 °C, and SG film annealed at 560 °C is important for the paper. Even if there are phases apart from Bi4Ge3O12 and Bi2GeO5, they should be mentioned in the figure and in the text. In case there are patterns that remain unidentified according to the reference patterns, they should be indicated in the caption as well. The demonstration styles used for XRD in the papers about bismuth-based oxide materials in the links below can be taken as examples:

https://www.hindawi.com/journals/acmp/2014/968349/

https://pubs.rsc.org/en/content/articlehtml/2017/ra/c7ra04375a

Answer:

All main peaks coming from Bi4Ge3O12 and Bi2GeO5 are marked in the theoretical spectra (see first answer) while the unknown were marked as “U”. An explanation was given in the text. These peaks are presented as in the references suggested by the Reviewer. Thank you.

Reviewer 2.

o “SG annealed at 560 °C” and “SG film annealed at 560 °C” are quite similar naming styles. In order to make it clearer, “SG annealed at 560 °C” can be changed as “SG powder annealed at 560 °C”.

Answer:

The SG term was changed with “sol-gel” in the text and figures, while the “SG annealed at 560 °C” was changed as “sol-gel powder annealed at 560 °C” or “sol-gel powder initial”.

Reviewer 2.

o It is not clear which plane belongs to which XRD peak for too closely located peaks with close intensity values in the figure. Putting an arrow between the peak and the (hkl) plane to make it more distinguishable should be helpful in making the demonstration clearer. On the figure, (310) is written as [3,1,1] as a mistake. The numerical mistake should also be corrected.

o In order for the manuscript to look more consistent, a parallel demonstration in the naming of the peaks with diffraction planes should be followed both for Figure 2a and Figure 2b.

Answer:

All the (hkl) planes are converted accordingly with the reviewer suggestion but only for the theoretical XRD patterns, while the experimental peaks are marked as BGO or Bi2GeO5 in correspondence with the theoretical one.

Reviewer 2.

o The meaning of the “*” symbol used for particular peaks of “SG initial” pattern is not indicated neither in the figure nor in the text. If it is to be used in the updated version as well, what it stands for should be clearly stated in the captions.

Answer:

The term “*” was removed for clarity. The figures 2 a) and b) were updated.

Reviewer 2.

ï‚· Figure 3 and 4 are blurry. It would be better in case they are presented in a more comprehensible way.

Answer:

Figures 3 and 4 were now prepared in Origin 9.0 for a better view. The previous figures were taken from the original SEM data.

Reviewer 2.

ï‚· Figure 8 is not mentioned in the text. It just takes place in the caption. It should be included in the text.

Answer:

Figure 8 is an image of the Raman microscope and was mentioned in the last phrase before Conclusions.

Reviewer 2.

ï‚· The entire text should be carefully improved because notation mistakes like writing “C” instead of “°C” or overlooking subscripts in compound formulas are common.

Answer:

The Celsius degree “C” was changed with “0Caccordingly.

Small changes, like those suggested by the Reviewers, are not highlighted but when the text was added this was highlighted in red.

Round 2

Reviewer 1 Report

Authors have improved the contents of the manuscript. I would recommend to make more wider and higher picture of XRD spectrum so that spectrum of sol-gel powder initial would be more clear. Also, in Figure 3 and 4,

X-scale will be - Energy ( KeV) and Y-scale is Intensity (cps/ev).

I would recommend its publication after these minor revisions.

Author Response

Reviewer 1

Authors have improved the contents of the manuscript. I would recommend to make more wider and higher picture of XRD spectrum so that spectrum of sol-gel powder initial would be more clear. Also, in Figure 3 and 4, X-scale will be - Energy ( KeV) and Y-scale is Intensity (cps/ev).

I would recommend its publication after these minor revisions.

Answer:

All pictures of XRD (figures 2a) and 2b)) but also Figures 3 and 4 were enlarged and the "sol-gel powder initial" was dragged about 6 times for a better view.

The axes from Figures 3 and 4 were corrected in Energy and Intensity. Thank you.

Anyhow, all figures are in tiff format at high resolution in the zip file sent to the Editor.

Reviewer 2 Report

The authors have carefully and appropriately implemented the required changes. There are only minor points left related to Figure 2a and 2b which remain to be corrected:

The XRD pattern which belongs to "sol-gel powder initial", plotted in dark blue, is not completely visible. To reveal the entire pattern, the intensity axis should be further dragged in the -y direction. Additionally, both figures should be broadened, it would improve the clarity of the content of the images.

Author Response

Reviewer 2

The authors have carefully and appropriately implemented the required changes. There are only minor points left related to Figure 2a and 2b which remain to be corrected:

The XRD pattern which belongs to "sol-gel powder initial", plotted in dark blue, is not completely visible. To reveal the entire pattern, the intensity axis should be further dragged in the -y direction. Additionally, both figures should be broadened, it would improve the clarity of the content of the images.

Answer:

The "sol-gel powder initial", plotted in blue was dragged about 6 times for a better view. Consecutively, the other spectra were shifted on the Oy scale and some peaks were added, both in the experimental and theoretical spectra. Thank you.

Anyhow, all figures are in tiff format at high resolution in the zip file sent to the Editor.